# Adaptive Algorithms for Batteryless LoRa-Based Sensors

**DOI:** 10.3390/s23146568

**Published:** 2023-07-21

**Authors:** Fabrizio Giuliano, Antonino Pagano, Daniele Croce, Gianpaolo Vitale, Ilenia Tinnirello

**Affiliations:** 1Department of Engineering, University of Palermo, 90128 Palermo, Italy; antonino.pagano@unipa.it (A.P.); daniele.croce@unipa.it (D.C.); ilenia.tinnirello@unipa.it (I.T.); 2Palermo Research Unit, CNIT (National Inter-University Consortium for Telecommunications), 90128 Palermo, Italy; 3Institute for High Performance Computing and Networking, National Research Council (CNR), 90146 Palermo, Italy; gianpaolo.vitale@icar.cnr.it; 4Department of Electrical, Electronics and Computer Science Engineering, University of Catania, 95131 Catania, Italy

**Keywords:** adaptive algorithms, batteryless, energy harvesting, internet of things, LoRa, wireless sensor networks

## Abstract

Ambient energy-powered sensors are becoming increasingly crucial for the sustainability of the Internet-of-Things (IoT). In particular, batteryless sensors are a cost-effective solution that require no battery maintenance, last longer and have greater weatherproofing properties due to the lack of a battery access panel. In this work, we study adaptive transmission algorithms to improve the performance of batteryless IoT sensors based on the LoRa protocol. First, we characterize the device power consumption during sensor measurement and/or transmission events. Then, we consider different scenarios and dynamically tune the most critical network parameters, such as inter-packet transmission time, data redundancy and packet size, to optimize the operation of the device. We design appropriate capacity-based storage, considering a renewable energy source (e.g., photovoltaic panel), and we analyze the probability of energy failures by exploiting both theoretical models and real energy traces. The results can be used as feedback to re-design the device to have an appropriate amount energy storage and meet certain reliability constraints. Finally, a cost analysis is also provided for the energy characteristics of our system, taking into account the dimensioning of both the capacitor and solar panel.

## 1. Introduction

The rise of the Internet-of-Things’ (IoT’s) demands for efficient and sustainable power sources and, therefore, ambient energy-powered sensors have attracted significant attention due to their environmental and economic benefits. By capturing, converting, and storing energy from the immediate environment (such as solar radiation, thermal gradients, and mechanical vibrations), autonomous sensors are opening new perspectives in different fields, especially where power supply from the grid is absent, e.g., in agriculture [1,2]. Moreover, new batteryless sensors offer numerous advantages compared to conventional alternatives, making them an attractive choice for IoT applications. Indeed, a batteryless design allows for IoT devices to be powered with improved cost-effectiveness, recyclability, longevity, and weatherproofing due to the elimination of batteries. batteryless sensors find application in a wide range of IoT scenarios, such as agriculture (e.g., soil monitoring systems), healthcare (e.g., wearable or intra-body devices), and smart cities (e.g., solar-powered streetlights and traffic signals, surveillance cameras, etc.). As IoT technologies are becoming increasingly integrated into daily life, adopting batteryless sensors can significantly extend device lifespan, reduce carbon footprints and contribute to a more environmentally conscious society.

In general, an energy-autonomous system combines energy harvesting (EH) equipment with an energy storage system to ensure the continuity of the supply. The total cost depends on the level of reliability required: a lower probability of failure results in a higher cost. The device sensors and the data transmission system represent a variable load whose supply must be provided by the harvesting and storage sub-system [3,4]. Indeed, the energy autonomy requirement implies the correct sizing and management of these three components, i.e., sources, load, and storage. On the one hand, the size of harvesting and storage cannot exceed certain limits (mainly due to cost and size issues); on the other hand, it is necessary to avoid energy failures and to ensure the transmission of sensor data [5]. Providing solutions to avoid the use of batteries by harvesting energy from the environment would encourage the deployment of IoT devices. The design of batteryless systems is therefore a complex task, which includes the correct sizing of both the harvesting and of the storage systems and the energy management optimization. Many energy sources can be considered in the design, e.g., solar cells, vibration-based systems, thermoelectric, and solar thermoelectric methods [6]. However, the highest power density (about 15 mW/cm${}^{2}$) is provided by solar cells [7] and, for this reason, in this paper, we focus mainly on this energy source. For energy storage, rechargeable batteries or supercapacitors are usually adopted. Batteries offer a higher energy density compared to supercapacitors; however, supercapacitors have very low internal impedance, allowing for a higher current pulse without detriments to efficiency [8]. In the charge/discharge process, energy flows twice in the same circuit path of the accumulator, and this suggests that we should move toward low-impedance storage systems, such as super-capacitors, to improve the efficiency of the storage system. Moreover, batteryless sensors are more cost-effective and recyclable, last longer, require no battery maintenance, and have greater weatherproofing due to the lack of a battery access panel [9]. These benefits motivate the present work. However, there are still several problems and hurdles that must be tackled. For instance, energy-harvesting power sources with low and intermittent output, energy storage capacitors, wireless interference, and intermittent random access transmissions can frequently cause power and/or communication failures.

In the literature, various energy-aware transmission algorithms have been proposed to optimize the performance of battery-free IoT sensors [10,11,12,13]. Adaptive transmission algorithms play a crucial role in maximizing the efficiency and reliability of these sensors [14,15]. The aim is to balance energy consumption with communication requirements, considering parameters such as transmission power, data rate, modulation scheme, and duty cycle [16,17]. By adapting the data transmission to the available energy and specific application needs, these algorithms improve the overall performance, reliability, and lifespan of battery-free IoT devices. For instance, in [10], it is possible to observe the implementation of specific wake-up policies through optimized algorithms for the best sampling frequency implementation based on the power received from the source. Furthermore, the use of adaptive algorithms can adress the challenge of incorporating intelligence into small battery-free sensors subject to the constraints of limited energy resources and dynamically adapting computation conditions based on the unpredictable nature of harvested energy [18,19,20]. Adaptive protocols have been proposed in [21] to promote the coexistence of different battery-free devices, with varying transmission requirements, and to dynamically allocate transmission slots for 40 devices without requiring prior knowledge of the environment. Furthermore, adaptive algorithms can decide whether to use the harvested energy immediately for transmission or to store it for future communications. This approach optimizes energy management and ensures efficient resource allocation, reducing, for example, transmission latency [12] and improving coexistence [21]. All this further highlights the importance of adaptive algorithms for battery-free IoT sensors.

In this paper, we analyzed two commercial LoRa devices: a FiPy module equipped with Pytrack expansion board and a TTGO T-Beam ESP32 board. These devices are based on the ESP32 System on Chip (SoC) and feature an SX1276 LoRa transceiver. We believe that an approach based on real measurements is more realistic than simulation-based studies. Moreover, to the best of our knowledge, we are the first to optimize specific network parameters for improved energy management, such as packet size or payload redundancy, in the context of batteryless LoRa devices. We thus propose a general approach that models both the energy source and storage, together with optimized energy management for the transmission system, which are tuned in real-time to comply with the available energy. Results are verified on a realistic case study, analyzing different performance requirements such as the loss of energy probability. Finally, we provide a cost analysis for dimensioning the energy storage and EH sub-systems. In summary, the main contributions of this paper are:An architectural design and energy consumption analysis of the batteryless device, based on two commercial LoRa transmitters;The optimization of data transmission using adaptive scheduling and redundancy schemes;The validation of our model with a real dataset, evaluating the probability of energy failure and cost analysis for energy storage and harvesting.

The rest of the paper is organized as follows: Section 2 presents state-of-art of batteryless IoT sensors, and Section 3 describes the general architecture of the proposed system and models the device energy consumption. Section 4 is devoted to energy harvesting and management, including generation and optimization issues, while results are presented in Section 5, varying storage size and analyzing the energy production in different seasons of the year. Finally, Section 6 discusses cost/performance issues, and Section 7 concludes the paper.

## 2. Background and Related Work

In this section, we present some background and literature works aimed at developing batteryless devices and optimizing transmission parameters based on the available energy from EH sources.

### 2.1. Batteryless Devices for IoT

Making a device completely energy-neutral requires a thorough analysis of power consumption in different working states [22]. Several works developed theoretical models for batteryless devices using emulated environments [22,23,24]. For example, Delgado et al. [22] provide a Markov model to characterize the performance of battery-free LoRaWAN devices for uplink and downlink (UL/DL) transmissions and assess their performance in terms of the model parameters (i.e., device configuration, application behavior, and environmental conditions). The study demonstrates that a 47 mF capacitor can handle 1 Byte SF7 transmissions every 60 s at a 1 mW energy harvesting rate. Indeed, the work shows that battery-free LoRaWAN communications are possible with the correct setup (i.e., capacitor size and turn-on voltage threshold) for various application behaviors (i.e., transmission interval, packet sizes, energy harvesting rate). Furthermore, in [24], the effectiveness of battery-free LoRa networks powered by ambient EH sources has been studied, assuming random transmission schemes. By using methods from stochastic geometry and Markov chain analysis, a mathematical model for each of the system’s components was built, and the likelihood of an energy and communication outage was analytically computed. The study has shown that LoRa networks’ adaptive data rate (ADR) can result in energy outages when employing higher spreading factors, and suggests adaptive charging time schemes as a successful remedy. In [23], authors investigated the optimal parameters to schedule application tasks on batteryless IoT sensor devices. Using an environment emulator and a SODAQ ExpLoRer board, the authors validated a mathematical model for choosing the optimal parameters, in terms of minimum application cycle completion time, at which to perform sensing and transmission, considering different device and environmental conditions. The analysis shows that a device using LoRaWAN Class A, equipped with a capacitor of 10 mF, can measure the temperature and transmit its data at least once every 5 s, and can harvest at least 50 mW (10 mA of current). Finally, the work in [16] developed an energy-aware system model to operate battery-free IoT devices that include several wireless communication protocols. To assess the total energy efficiency of the IoT network, simulations based on a probabilistic sensing model are used. According to the results, to achieve self-sustainability in a heterogeneous short- and long-range network and enhance energy efficiency, an energy harvesting device combining a solar panel with a 270 F lithium-ion super-capacitor must be utilized as a power storage device.

However, these theoretical results, particularly [22,24], might be overly optimistic in terms of inter-arrival times between packets and the size of the storage capacitor, and require confirmation in real-world implementation. The studies [25,26] proposed some prototypes of batteryless nodes based on LoRa technology. Specifically, Orfei et al. [25] demonstrated the performance of a batteryless sensor for monitoring road traffic and bridge conditions, powered by a low-cost electromagnetic EH device, which employs an array of permanent magnets to improve energy efficiency. The collected energy is stored in a supercapacitor and powers an ARM Cortex M0+ microcontroller and a LoRa radio module to transmit information. On the other hand, Boitier et al. [26] introduced a self-contained LoRa sensor with a photovoltaic power source and a pair of 25 F supercapacitors for energy storage. This solution assures 11 days of storage life in the absence of light. The proposed system also includes a circuit for energy management and troubleshooting on the first startup.

### 2.2. Energy Management Optimization

One of the challenges for batteryless IoT devices is their limited energy availability and reliance on the surrounding environment. If a device runs out of energy, it cannot perform its functions until the harvesting system recharges its energy storage [27]. Generally, the main parameters used to optimize energy consumption on wireless nodes are the data transmission interval *T* (i.e., the time between two consecutive packets) and data overhead/redundancy NR (which can be implemented in several ways, e.g., through coding or transmission repetition). Therefore, developing a good policy for choosing the periodic interval to transmit information and tune the amount of data redundancy is of paramount importance. Due to the imperfect predictability of real-world events that affect energy sources, the optimization strategy must strike a balance between capturing these events and consuming all available energy. To achieve this, recent studies have suggested using artificial intelligence (AI) to learn such policies for battery-free sensors. In particular, it has been shown that data-driven strategies, such as reinforcement learning, can be exploited because the amount of available energy changes in similar patterns to other close environments. [17,28,29,30]. In other words, energy availability could be predicted to make better decisions when implementing proper energy resource management strategies in batteryless devices [14,31]. However, the unpredictable nature of the energy sources requires large datasets to train these AI-based systems and optimize the device parameters in relation to the energy storage process. Moreover, these strategies should be implementable on low-complexity hardware, which is usually different for AI algorithms.

In [15], several scheduling algorithms are applied to batteryless LoRaWAN nodes, analyzing their performance in various simulated scenarios. Based on real-world EH measurements gathered from a testbed, the work studies the impact of energy-aware schemes on the number of transmitted packets and the mean packet interval. The results demonstrate that energy-aware algorithms can significantly enhance the performance of batteryless LoRaWAN nodes. However, the presented results are strongly influenced by the harvesting capabilities of the nodes. The works [32,33] suggest a simple approach for network optimization, exploiting a revised sigmoid function that can be easily computed on low-cost hardware. The operating strategy can be adjusted based on a small subset of the most recent energy information or even the last two samples. Specifically, these methods reduce energy consumption by modifying the sample rate based on the remaining battery level [32] or the harvested ambient energy [33]. In particular, the authors in [33] developed an objective function to optimize the transmission period *T*, with dynamic sampling adaptation schemes that can be classified into two possible categories: (a) threshold-based sampling adaptation (T-ASA), which basically adapts the sampling time based on energy thresholds, or (b) data-driven adaptive sampling algorithm (DDASA) by exploiting a sigmoid function to dynamically adapt the sampling time. Moreover, this scheme can be adjusted to the variability of the ambient energy, requires less computational capacity compared to complex AI or Markov chain schemes, and can be easily implemented on ultra-low power boards, such as batteryless LoRa nodes [34]. However, both these studies consider the use of batteries to supply wireless sensors; the first one used a battery-powered sensor without EH capabilities [32], while the second one used a photovoltaic source and a battery as a storage system [33]. Conversely, this work investigates, for the first time, the possibility of using the revised sigmoid function algorithm to optimize the transmission parameters of a batteryless LoRa node powered by a solar panel.

In particular, we adopted DDASA to dynamically set the transmission time of the device and compute the optimal value with a low-complexity algorithm that can be implemented on low-cost hardware. Moreover, a sigmoid-based approach can also be applied to adjust the data redundancy according to the environmental conditions (e.g., state of charge or solar radiation). This consideration is also essential for the correct design of the nodes: indeed, from the literature, it is clear that an appropriate choice of electronic components, along with the correct optimization of transmission parameters (*T*, NR, packet size, etc.), is essential to achieve the right balance between energy and economic efficiency. For example, regarding the storage system capacity, very different values have been used in the state of the art, ranging from 10 mF [23] to 270 F [16]; for this reason, we believe it is necessary to design the system, taking into consideration not only transmission parameters and energy efficiency but also the costs of the entire system. To achieve this goal, our work considers the loss of energy probability (LoEP). Lastly, to the best of our knowledge, this study represents the first characterization of batteryless LoRaWAN sensors, powered by solar energy, using adaptive algorithms to optimize transmission parameters and compute the LoEP to minimize the costs of the sensor node architecture.

## 3. System Architecture

The main components of the proposed adaptation system are presented in Figure 1. The batteryless device acquires energy from a renewable energy source (e.g., solar energy) stored in a supercapacitor. Physical components, such as the solar panel and supercapacitor, must be designed to provide enough energy in normal operating conditions. This represents a significant challenge because the device design is critical to optimize the amount of energy harvested from renewable sources to perform sensing and transmission operations. Regarding data processing, when the device has acquired new data from the sensors and has enough energy, it transmits a new packet according to two parameters: (i) the transmission interval, i.e., the time *T* between two consecutive packets; (ii) the packet data size, with redundancy parameter NR.

In our implementation, these parameters are either fixed or tuned dynamically according to the algorithm described in Section 3.2, which considers the energy stored in the device.

As a renewable energy source, we used a photovoltaic (PV) panel as an energy source. In particular, we considered both a theoretical model and a real radiation data set as input, analyzing the impact of the EH source on the device and the effect of possible energy failures. Regarding the theoretical model, we used Duffie’s radiation model [35], which is detailed in Section 4.2. We employed the open data provided in [36,37] for the real radiation dataset.

### 3.1. Device Energy Model

The energy harvesting-based device was designed, taking the following parameters into account:Device consumption profile ($E_{DEV}$);Solar panel size ($S_{PV}$);Energy storage capacity (*C*);Transmission interval (*T*);Data redundancy (NR).

For simplicity, the proposed model considers discrete time samples $n=k\cdot T_{PV},\;k\in N$, where $T_{PV}$ is the energy source measurement sampling time, e.g., 1 h or 15 min. Without loss of generality, in our scenario, we take into account a PV panel as energy source, which can be modeled as:(1)$$E_{PV}\left(n\right)=\eta \cdot S_{PV}\cdot G_{solar}\left(n\right)$$
where η represents the PV efficiency, $S_{PV}$ is the panel size and $G_{solar}\left(n\right)$ is the solar radiation in the time interval between [n−1,n], i.e., $\left[k-1,k\right]\cdot T_{PV}$. The efficiency η allows for different materials to be considered for the PV cell (mono, polycristalline or others).

The energy stored in the system is modeled using the following equation:(2)$$E_{ST}\left(n\right)=\min\left[E_{PV}\left(n\right)+E_{ST}\left(n-1\right)-E_{DEV}\cdot f_{TX}\left(n\right),E_{max}\right]$$
where $E_{ST}\left(n-1\right)$ is the energy stored by the system at time n−1, $E_{PV}\left(n\right)$ is the energy provided by the source, $E_{DEV}$ is the energy consumed by the device and $f_{TX}\left(n\right)$ is a function that is equal to 1 if there is a transmission in the last interval (zero otherwise), as formalized in Equation (Equation 4). The system energy storage is mainly composed of a supercapacitor which allows for high-speed charge accumulation, up to the $E_{max}$ limit (function of *C*). In our analysis, we consider that the charging time of the capacitor is very short compared to $T_{PV}$, so we can neglect the capacitor transient times. Finally, the energy consumed by the device $E_{DEV}$ is computed as the sum of three main components as:(3)$$E_{DEV}=E_{SENS}+E_{TX}\left(NR\right)+E_{RX}$$
where $E_{SENS}$ is the energy required to acquire and process data from the different sensors, $E_{TX}\left(NR\right)$ is the energy spent by the device to transmit a packet over the air and $E_{RX}$ is the energy used to perform the receiving operation.

Note that $E_{TX}$ depends not only on the parameters of the LoRa transmission protocol but also on the payload size. Such a payload can change dynamically to increase/decrease the number of sensor data sent in a single frame or to implement an additional redundancy mechanism to improve the reliability of LoRaWAN transmissions (tuned by the overall NR parameter). A simple way to implement this solution consists of introducing temporal redundancy, i.e., retransmitting the same measurement data into multiple transmissions, thus increasing the probability that at least one of the packets is received. This approach is already being exploited in commercial applications such as the Sensing Labs platform [38]. Thus, we implemented a sliding window mechanism that transmits the last NR measurements, with higher NR values yielding a greater probability of success. Figure 2 illustrates the shift-memory structure of NR, which updates the values whenever a new sensor measurement is performed.

Finally, the device can only transmit if $E_{ST}$ is high enough to perform a complete packet transmission. We model this functionality using the following activation function: (4)$$f_{TX}\left(n,T\right)=\left\{\begin{array}{c} 1\;\;\;\;E_{PV}\left(n\right)+E_{ST}\left(n-1\right)\geq E_{DEV}\;\;\wedge \;\;n=k\cdot T\\ 0\;\;\;\;otherwise \end{array}\right.$$
where the time interval *T* is the interval between two consecutive transmissions (for simplicity, we assume *n* to be a multiple of *T*). Clearly, the transmission delay decreases reductions in *T*; in contrast, the amount of energy needed for the transmissions increases (and vice-versa). Next, we discuss how to tune *T* and NR parameters based on the energy received from the renewable source.

### 3.2. DDASA-Based Transmission Algorithm

We defined a DDASA-based algorithm to dynamically adapt the transmission period of the sensor data. Indeed, as discussed in Section 2, the DDASA algorithm [33] can be employed to adapt sampling and data transmission to optimize the resource utilization of the device. In particular, we considered this algorithm to increase or reduce the transmission interval and the data redundancy according to the amount of energy available to optimize energy consumption and minimize transmission failures. The adaptation is represented by a revised sigmoid function, which is expressed by $y\left(x\right)=\frac{2}{1+e^{-x}}$, where x, in our case, is the difference in stored energy computed between two consecutive transmissions.

Figure 3 reports a flowchart of the proposed algorithm, while the pseudocode is described in Algorithm 1. The device checks whether the elapsed time $m\cdot T_{PV}$ is greater than the transmission period *T* and if the stored energy $E_{ST}$ is greater than the energy required $E_{DEV}$ to transmit a LoRa frame. A transmission occurs if these conditions are verified, after which the parameters *T* and NR are updated. The algorithm dynamically updates *T* and NR by considering two components: the variation in stored energy (line 8) and the gap between the stored energy s and the energy device (line 10). Moreover, the algorithm introduces two sigmoid parameters: $v_{1}$, which considers the accumulated energy variation between two transmissions, and $v_{2}$, which measures the gap between the current stored energy and the energy needed to process and transmit a data packet. Finally, the values of *T* and NR are constrained between $T_{min}=1$ h and $T_{max}=24$ h, while NR can assume values between 1 and 10.
**Algorithm 1** DDASA-based Transmission Algorithm$T=T_{min}$                                                                      ▹ Initialize variables$NR=NR_{max}$m=0n=0**while** true **do**    **if** $\left(m\cdot T_{PV}\geq T\right)\left(E_{ST}\left(n\right)\geq E_{DEV}\right)$ **then**        run(TxLoRaFrame)                                                    ▹ Transmit data        $b_{1}=\left(E_{ST}\left(n\right)-E_{ST}\left(n-m\right)\right)/mean\left(E_{ST}\right)$            ▹ Energy variation        m=0                                                                  ▹ reset time counter        $b_{2}=\left(E_{ST}\left(n\right)-E_{DEV}\right)/E_{max}$                                         ▹ energy gap        $v_{1}=\frac{2}{1+e^{-b_{1}}}$                                                         ▹ Sigmoid function 1        $v_{2}=\frac{2}{1+e^{-b_{2}}}$                                                         ▹ Sigmoid function 2        $T=min(T_{max},max\left(T_{min},\frac{T}{v_{1}\cdot v_{2}}\right))$                                    ▹ Update T        $NR=min(NR_{max},max\left(NR_{min},NR\cdot v_{1}\cdot v_{2}\right))$            ▹ Update NR    **end if**    m++    n++**end while**

## 4. Power Consumption Measurements and Solar Energy Model

In this section, we analyze the consumption of real LoRaWAN nodes for use in the adaptive algorithm described above. We used two commercial development boards, the LoRa FiPy and TTGO nodes. Additionally, we present a model to characterize solar energy radiation and its variability, which is crucial for our analysis. We compare the model output against a real dataset and discuss the design of the solar panel, characterizing the maximum power point for efficient energy conversion. This characterization allows for optimization of the proposed system, with an operating point that reduces the system sensitivity to the current fluctuations.

### 4.1. Device Consumption Measurements

In our analysis, we conducted experiments to evaluate the current consumption in four different operating modes (sensing, transmitting, receiving, and sleeping). As shown in Figure 4, the experimental setup was composed of a PV panel, a protection circuit, a supercapacitor and the LoRa device, with transmission settings summarized in Table 1. In particular, we evaluated the power consumption of the LoRa FiPy and TTGO nodes, which include the ESP32 SoC and the SX1276 LoRa transceiver. We used a Tektronix MSO 2024B oscilloscope with TCP0020 current probe to measure the PV Panel Voltage, source voltage (from the supercapacitor) and the current absorbed by the device. For example, the power consumption of the TTGO device is depicted in Figure 5 during different node activities. Specifically, the figure shows the device consumption during sensing, the transmission of a LoRa packet (LoRa TX), and the subsequent receive windows (LoRa RX1 and RX2) when the device listens for responses, acknowledgments, or downlink messages from the network server.

The average power consumption of the two considered devices in all the four mentioned states is outlined in Table 2. In particular, the table shows that, in the three active states (transmit, receive, sensing), the power consumption of the TTGO device is lower than the FiPy device. In contrast, the TTGO device initially had higher power consumption in the sleep state (about 10 mW), which was reduced at 0.15 mW by making the hardware changes suggested in [39]. Based on both devices’ energy consumption characterization outcomes, it was decided to proceed with the analysis focusing solely on the TTGO device, as it shows lower energy consumption in all the measured conditions.

### 4.2. Solar Energy Production Model

In this subsection, we model the solar energy production based on the radiation theoretical model explained in Chapter 1 of [40]. In particular, solar radiation can be predicted based on the day of the year, latitude, and azimuth. Energy reaches the Earth, is partially scattered by the atmosphere, and can be converted by photovoltaic devices into electrical energy. Generally, the solar radiation overall captured by a solar panel comprises three components: direct, diffuse, and reflected. In our analysis, we selected the correlations by Erbs et al. [35]: based on measurements taken at various locations in the United States, an experimental data regression model is proposed for the luminosity index to estimate the diffused solar radiation.

The next step involves estimating the power generated by a solar panel to predict hourly production and determine the energy flows exchanged with the LoRa device and supercapacitor. The power delivered from the photovoltaic panel was calculated considering the photoelectric conversion efficiency, panel area, solar radiation incident on the panel, and operating temperature, using the empirical relationship expressed in [41]. We analyzed two different radiation datasets to validate the theoretical model and measure the performance of the proposed adaptation algorithm. Thus, we compared the average monthly power output of the theoretical model against the real dataset provided in [36,37], at two reference locations (latitudes 38.132° and 45.45° N). These datasets are provided by the National Solar Radiation Database (NSRDB) of the USA and Servizio Informativo Agrometeorologico Siciliano (SIAS), Italy, respectively. The first dataset is a comprehensive and publicly available source of solar radiation and meteorological data offered by the National Renewable Energy Laboratory (NREL) in the United States. The NSRDB provides high-resolution solar irradiance data and covers an extensive period, from 1991 to the present, with regular updates. The NSRDB offers high temporal (hourly or sub-hourly) and spatial (4 km × 4 km) resolution data, making it suitable for detailed analyses and accurate energy production simulations. The dataset includes various solar radiation parameters such as global horizontal irradiance (GHI), direct normal irradiance (DNI), and diffuse horizontal irradiance (DHI). Additionally, it provides essential meteorological parameters such as temperature, humidity, wind speed, and precipitation. The NSRDB combines measurements from ground-based stations, satellite-derived data, and advanced atmospheric models to ensure accuracy and reliability. In particular, the spectral on-demand data service provides solar irradiances on inclined PV panels and emploies the Fast All-sky Radiation Model for Solar applications with Narrowband Irradiances on Tilted surfaces (FARMS-NIT) [42], which is a radiative transfer model developed at NREL specifically for the calculation of solar energy distribution in narrow-wavelength bands over inclined PV panels. When clear-sky conditions are present, the Simple Model of the Atmospheric Radiative Transfer of Sunshine (SMARTS) [43] is employed to determine the optical properties of the atmosphere. This model considers three paths of photon transmission and solves the radiative transfer equation using the single-scattering approximation to compute clear-sky radiances in narrow-wavelength bands. For cloudy-sky conditions, instead, FARMS-NIT utilizes the cloud reflectance of irradiance and bidirectional transmittance distribution function (BTDF) from a precomputed lookup table generated by the LibRadtran model with a 32-stream Discrete Ordinates Radiative Transfer (DISORT). By combining the cloud reflectance, BTDF, and clear-sky properties, FARMS-NIT computes spectral radiances on the land surface and plane-of-array (POA) irradiances. The NSRDB database is freely accessible through NREL’s online platform or APIs, allowing users to download data for specific locations, time periods, and parameters.

Regarding the second dataset provided by SIAS (an agrometeorological information service in Sicily, Italy), it is based on a real-time weather monitoring system employing a network of meteorological stations distributed across the island of Sicily. The stations have wind speed sensors at 2 m and 10 m height, utilizing Robinson cups and optoelectronic transducer technology. The wind direction sensors at the same heights employ vane and optoelectronic transducer technology. Other sensors include a global radiation sensor (measuring cumulative solar radiation), air temperature sensor, relative humidity sensor, precipitation sensor (tipping bucket rain gauge), leaf wetness sensor, atmospheric pressure sensor, and snow depth sensor. The stations are synchronized in time to align with the forecasting models, and a weekly time check ensures station accuracy. The datalogger used is the MTX WST1800 [44] model, featuring a single-board CMOS microprocessor with 128 KB of RAM and 64 KB of EPROM memory. The SIAS dataset provides data acquired since 2003 at an hourly time resolution.

Comparing the theoretical model with the real datasets, Figure 6 reveals that, for both the considered locations, the theoretical model closely follows the actual solar radiation in the autumn and spring seasons; conversely, it tends toward underestimation in the summer months and overestimation during winters. The theoretical model performance is summarized in Table 3, in terms of Coefficient of Determination $\left(R^{2}\right)$, Root Mean Square Error (RMSE), and Mean Bias Error (MBE). In particular, the analysis of MBE is used to estimate the average bias of the model. The obtained results show low bias. According to RMSE, the model has a significant impact on outliers, and this is reasonable because the model does not account for unpredictable radiation fluctuations due to atmospheric events. Considering a maximum normal surface radiance of around 1000 W/m${}^{2}$ at sea level on a clear day, we can assume a normalized RMSE of approximately 10–13%. Finally, the $R^{2}$ values obtained are greater than 70%, which is an acceptable model fit.

### 4.3. Solar Panel Parameters

Maximizations in the efficiency of converted solar energy are strictly tied to solar panel selection. The power conversion efficiency (PCE) and maximum power point (MPP), as well as the size, are essential features for the design of the panel [45]. In our experiments, we employed a commercial silicon solar panel (sized 7.5 × 14 cm ). Therefore, a mathematical model, given in [46], was used to characterize the PV panel and identify the electrical parameters. In Figure 7, the PV panel’s I–V and P–V characteristics are reported, assuming 820 W/m${}^{2}$ of solar radiation. In the figure, $I_{sc}$, $V_{oc}$, $I_{mp}$, and $V_{mp}$ denote short-circuit current, open-circuit voltage, the maximum power point current, and the maximum power point voltage, respectively. By extracting these four parameters, we can identify the MPP operating condition in which the power transferred from the source to the load is maximized. Thanks to this characterization of the panel, the proposed dynamic algorithm can be tuned to work on the operating point closest to the $V_{mp}$ voltage, reducing the system’s sensitivity to current at the MPP point. From the figure, it is clear that the operating voltage value ($V_{ope}$) must fall within the supply voltage range of [3.3–5.2] V.

## 5. Evaluation and Results

In order to evaluate the proposed system, we exploited the consumption model obtained from the device characterization, as discussed in Section 4, to simulate the behavior of the system by changing design parameters such as capacity and solar panel size. We also employed the adaptation algorithm, based on energy-aware DDASA, to dynamically adapt the transmission period *T* and payload size as a function of NR, as described in Algorithm 1. Finally, we conducted simulated experiments for one year using both the energy radiation model and real-world radiation datasets, as described in Section 4.2.

### 5.1. Performance of the Adaptation Algorithm

As a first experiment, we validated the adaptation algorithm capabilities to optimize the transmission parameters (i.e., transmission interval and NR). In particular, we measured the number of correct transmissions (#TX_DONE), the number of transmissions that failed due to insufficient energy (#TX_FAILED), and the number of bytes transmitted per packet. A transmission is considered failed when the stored energy is not sufficient to perform the overall procedure of sensing, processing, and transmission. Moreover, we compute the Loss of Energy Probability (LoEP), which directly translates to a packet loss, as $\frac{\#TX_{failed}}{\#TX_{failed}+\#TX_{done}}$. We analyze the impact of data redundancy defining a Loss of Information Probability (LoIP), i.e., the probability of losing NR consecutive packets, which can be computed as $LoIP=LoEP^{NR}$.

Table 4 and Table 5 summarize the results obtained using the theoretical model or the real dataset, respectively. For the experiments, we consider C=4.5 F and PV=0.01 m${}^{2}$, and we assume an EH efficiency of 15 mW/cm${}^{2}$ [7] and a latitude of 38.132°. The tables report the average TX interval *T* and NR parameters computed by the adaptive algorithm, the LoEP and LoIP probabilities, and the average packet size, which is directly influenced by NR and the number of sensors on the device (we assume a total sensor data of 2 Bytes, i.e., 2·NR bytes). Values show that during higher solar radiation periods (mainly summer and spring), the adaptation algorithm lowers *T* and increases NR, obtaining a good number of successful transmissions and resulting in a low LoEP compared to other seasons with lower radiation intensity (autumn and winter). For example, 1167 successful transmissions were obtained during the summer season using the theoretical radiation model while, in the winter, only 691 frames were successfully transmitted. Similar numbers were obtained using the real data. Note that, with the specific values selected for C and PV size, the LoEP for the theoretical model is generally lower than the real dataset. This is due to abrupt weather events (e.g., cloud obfuscation), which are not included in the theoretical model, but are recorded in the real dataset. In any case, the tables show that LoIP is extremely low, even with LoEP as high as 20% (winter and autumn seasons in Table 5). This demonstrates the importance of the data redundancy NR, which helps recover lost packets by retransmitting data multiple times.

Figure 8 shows a Cumulative Distribution Function (CDF) representation of the algorithm execution time and the CDF of the time deviation from the scheduled transmission. Figure 8a shows that the running time of the algorithm on the hardware used in our lab, while Figure 8b demonstrates the algorithm’s bility of to cope with unexpected delays (due to power failures), measuring the deviation from the scheduled transmission time. Regarding the processing time, this includes the time elapsed to perform the algorithm computation, the conditional operations, and the execution of the radio commands. From Figure 8a, computed over 1 year and roughly 3000 transmission events, it is clear that the system usually requires less than 50 µs to perform the algorithm operations, with a median value of 12 µs. Finally, Figure 8b shows the transmission deviation, defined as the time between the scheduled transmission event and the real packet transmission (which might be delayed due to power failure). The figure shows that around 98% of transmission attempts are executed without any delay, demonstrating the accuracy of the algorithm.

### 5.2. Impact of Supercapacitor and PV Panel Size

A second set of experiments was conducted by varying the values of *C* and PV and analyzing the overall LoEP obtained using the proposed adaptive algorithm. The results are summarized in Figure 9 and Figure 10, considering a one-year observation period for two different latitudes. The heatmap shown in the figures depicts the average LoEP as a function of the capacity and solar panel size, respectively, on the x- and y-axes. Moreover, we highlight values that are less than 5% of LoEP by explicitly indicating the obtained value in the plot. These figures are helpful for establishing some component constraints for the optimal design of the device, as discussed in the following section.

## 6. Design Parameters and Cost Analysis

Fixing a LoEP (or LoIP) threshold involves the definition of acceptable loss probability and, in the design of a batteryless device, can help identify appropriate design parameters (such as capacity and solar panel size) that achieve this threshold value. Together with these parameters, the design optimization should also consider other factors, such as cost. Concerning Figure 9 and Figure 10, values of C and PV can be identified so as to satisfy the constraints on the LoEP (e.g., less than 5%); in this way, the choice of components for given LoEP can be made by optimizing the cost. However, the optimal choice of source and storage size must take into account the availability on the market. The required values can be obtained directly by a single unit or by composing multiple smaller components until the desired value is reached. In particular, concerning the size of the PV source, the choice can start from a single cell, but array can also be considered. Similarly, the capacitance of the storage system can be composed by the parallel connection of two or more capacitors, if necessary. It should be noted that the total cost usually does not rise linearly; indeed, the market often proposes cheaper components, with higher performances for the corresponding elementary units that are to be connected.

We examined three potential commercial solutions for supercapacitors and solar panels to emphasize this phenomenon. Their characteristics and costs are displayed in Table 6 and Table 7, respectively. Then, we assessed all possible combinations of supercapacitors and photovoltaic cells to select the most cost-effective solution that maintained a LoEP of 5% or less, focusing on a Duffie’s radiation model with a specific location of latitude 38.132°.

Specifically, the combinations, analyzed with their costs, are shown in Table 8 and Table 9, where the most cost-effective solutions have been underlined. Regarding supercapacitors, the cost analysis in Table 8 shows that, for capacitance values required in the range (3–5.5) F, the most economical solution is provided by the adoption of two units “B” supercapacitors in parallel, obtaining a total capacitance of 5 F at the cost of €4.46. On the contrary, for desired capacitance values between 6 F and 7.5 F, choosing a single “C”-type supercapacitor of 7.5 F at the cost of €4.75 is more convenient. Similar considerations hold for the photovoltaic sources detailed in Table 9. In fact, for PV areas of 0.025 m${}^{2}$ or less, choosing a single element of type “B”, of size 0.026 m${}^{2}$, at a cost of €16 is the cheapest solution. For an area between 0.030 and 0.035 m${}^{2}$, it would be more convenient to adopt several elementary “A”-type cells in series. For the remaining desired surface area values, listed in Table 9, a pair in series of “B”-type components should be adopted.

Once the optimal system design has been obtained, the LoEP and LoIP can be re-analyzed with these new values. In particular, in the last experiment, we measured the loss probabilities when varying *T* and NR (in particular, by tuning $T_{max}$ and $NR_{max}$), with a fixed C=5 F and $S_{PV}=0.035$ m${}^{2}$. Figure 11 shows the performance of the adaptation algorithm in terms of LoEP, as a function of $T_{max}$. The figure clearly shows that the LoEP limit of 5% is satisfied when $T_{max}$ is greater than 5 h. Instead, with a $T_{max}$ fixed to 5 h, Figure 12 shows the LoIP when varying $NR_{max}$ between 1 and 10. In this case, we can highlight an exponential reduction in information loss probability, which can ensure data transmission even with small values of NR.

## 7. Conclusions

In this work, we analyzed LoRa-based batteryless devices, optimizing the transmission parameters through an energy-aware adaptation algorithm. By characterizing and modeling the power consumption of two batteryless sensors and using theoretical energy models and real radiation traces, we studied how to dynamically tune the transmission interval and payload size to cope with the available energy. Finally, we provided a general approach for the design of capacity-based storage and PV panel size, obtaining a cost-effective methodology to design batteryless solutions for ambient-energy-powered LoRa sensors. The results showed that, for a given threshold probability of power failure, the proposed approach can successfully optimize the device’s energy consumption by automatically setting the relevant transmission parameters. Moreover, information loss can be dramatically reduced simply by repeating the data transmission in multiple packets. From the presented results, a system composed of a 5 F supercapacitor and a PV panel of 0.035 m${}^{2}$ is capable of transmitting data packets every 5 h, with a redundancy of 3 and a LoIP of 10${}^{-2}$. Finally, the LoEP results obtained with the proposed algorithm, combined with an in-depth cost analysis, has allowed for the most economical solution to be selected for the dimensioning of the PV panel and supercapacitor, balancing implementation costs and energy failure probability. The proposed approach is applicable to different IoT applications that require autonomous energy systems, such as agriculture or emergency scenarios. In future research activities, we will test the proposed algorithm on a large-scale deployment and study the impact of networking and modulation parameters, such as the resource allocation of different spreading factors and adaptive data rate algorithms.

## Figures and Tables

**Figure 1 sensors-23-06568-f001:**
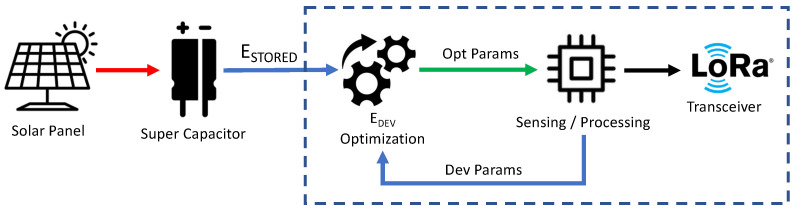
System Architecture and logical components.

**Figure 2 sensors-23-06568-f002:**
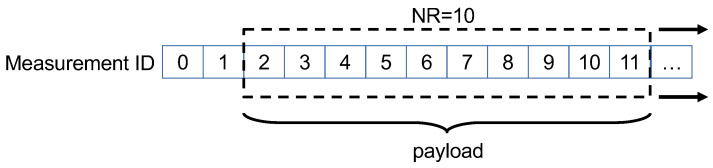
Sliding window scheme for data redundancy NR.

**Figure 3 sensors-23-06568-f003:**
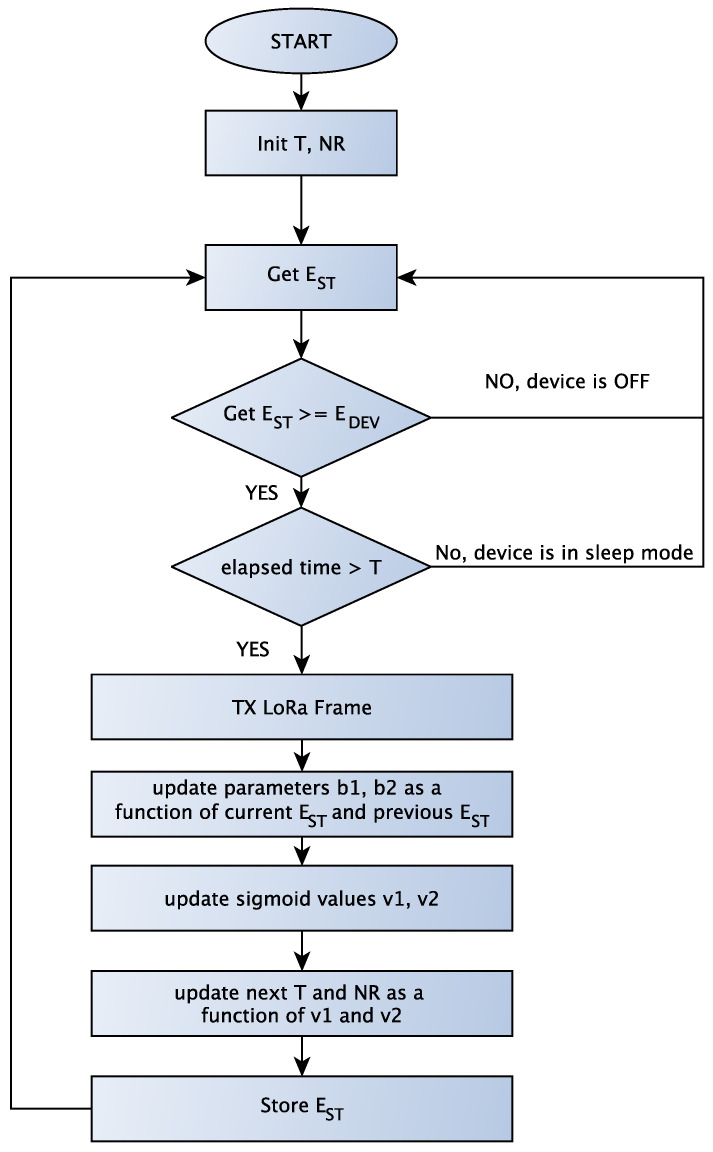
Flowchart of the DDASA-based Transmission Algorithm.

**Figure 4 sensors-23-06568-f004:**
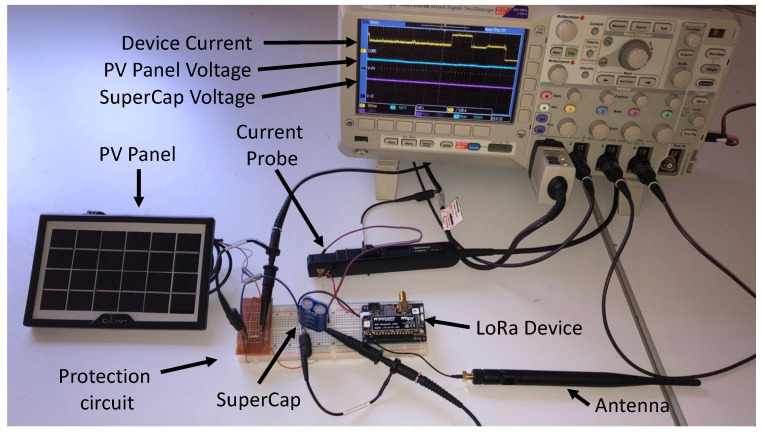
Experimental testbed setup used to measure the power consumption of the devices.

**Figure 5 sensors-23-06568-f005:**
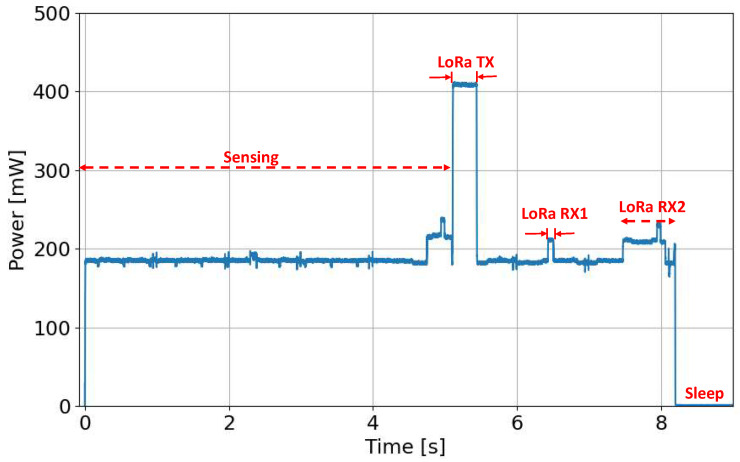
TTGO node power consumption in different working states.

**Figure 6 sensors-23-06568-f006:**
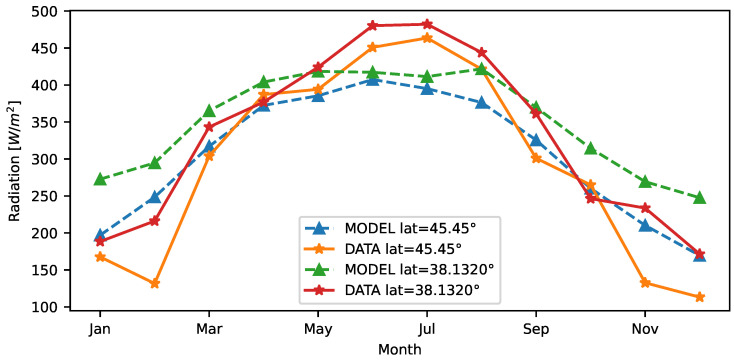
Monthly average solar radiation of the theoretical model and real dataset.

**Figure 7 sensors-23-06568-f007:**
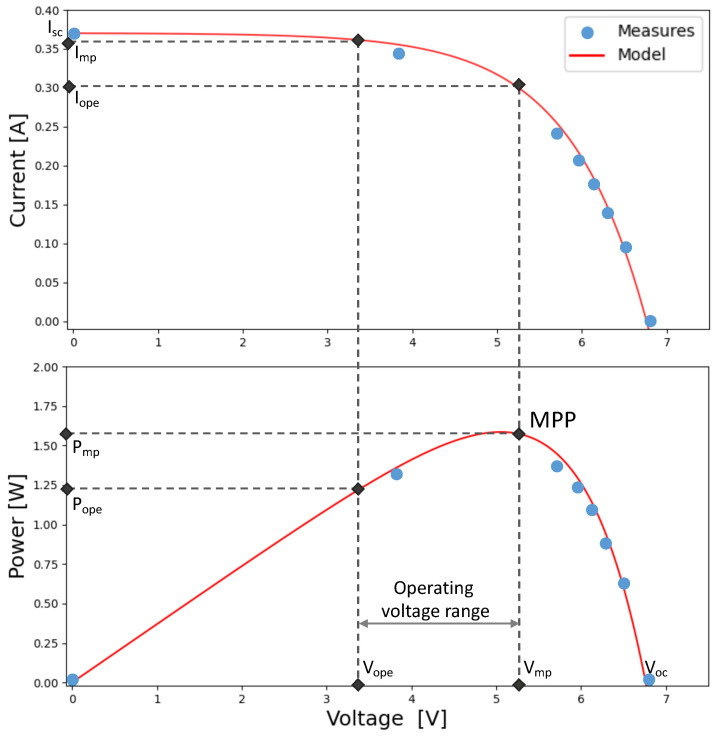
Solar panel characterization under 820 W/m${}^{2}$ of solar exposure.

**Figure 8 sensors-23-06568-f008:**
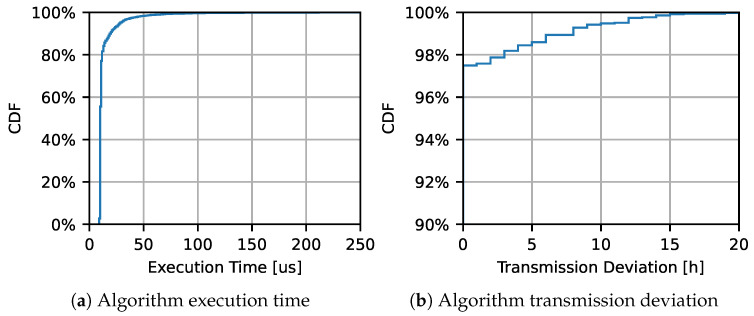
Complexity and accuracy of the proposed algorithm. CDF of the execution time and of the time deviation from the planned transmission time due to power failures.

**Figure 9 sensors-23-06568-f009:**
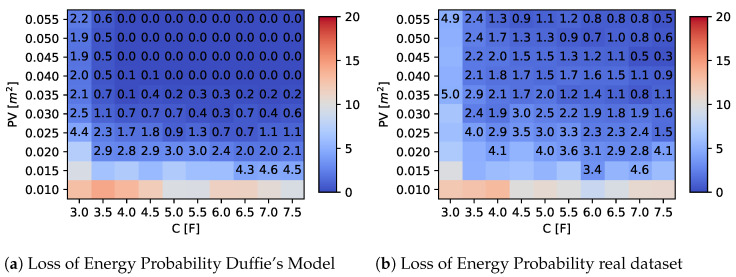
LoEP results at latitude = 38.132°.

**Figure 10 sensors-23-06568-f010:**
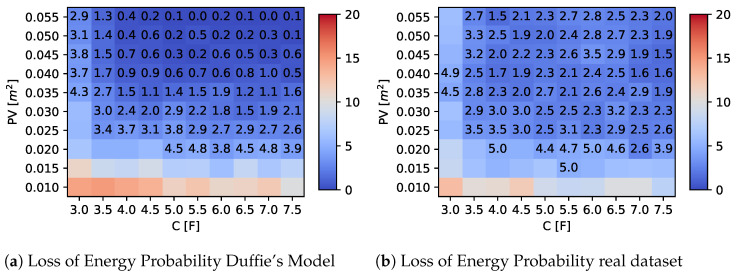
LoEP results at latitude = 45.45°.

**Figure 11 sensors-23-06568-f011:**
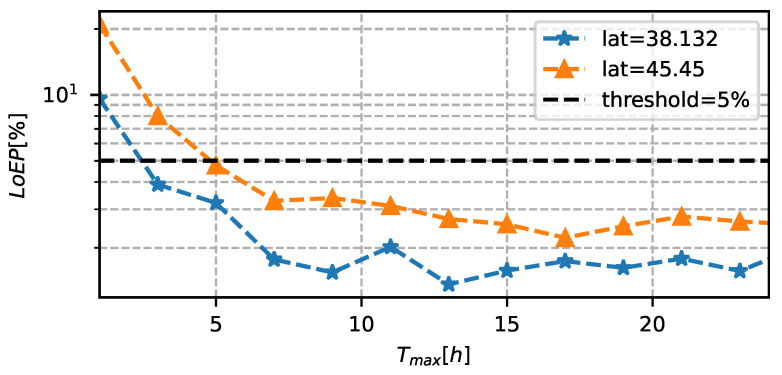
LoEP as a function of $T_{max}$, with C=5 F, $S_{PV}=0.035\;$
$m^{2}$, $NR_{max}=10$.

**Figure 12 sensors-23-06568-f012:**
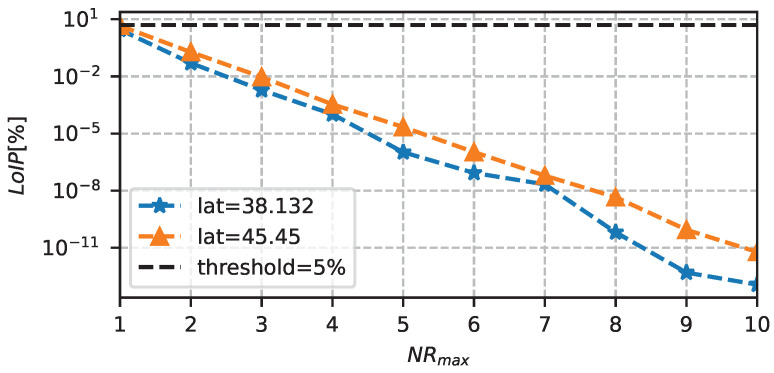
LoIP as a function of $NR_{max}$, with C=5 F, $S_{PV}=0.035\;$
$m^{2}$, $T_{max}=5$ h.

**Table 1 sensors-23-06568-t001:** Default values used for energy consumption measurements.

Setting	Value
Supply Voltage	3.7 V
Frequency	channel hopping in 868 MHz band
Spreading Factor	12
Forward Error Correction	4/5
CRC	Enabled
Payload length	2 bytes
Preamble Length	8 symbols

**Table 2 sensors-23-06568-t002:** Power consumption of the two devices in the different operating modes.

Device	Transmit	Receive	Sensing	Sleep
FiPy	820 mW	600 mW	560 mW	0.225 mW
TTGO	420 mW	228 mW	195 mW	0.15 mW

**Table 3 sensors-23-06568-t003:** Difference between the theoretical model and real dataset at two latitudes.

Latitude	$R^{2}$	RMSE [W/m${}^{2}$]	MBE [W/m${}^{2}$]
45.45°	0.71	135.57	−1.63
38.132°	0.83	109.03	1.01

**Table 4 sensors-23-06568-t004:** Results obtained per season using the theoretical model with C = 4.5 F and PV = 0.01 m${}^{2}$.

Season	Average *T* [h]	Average NR	#TX_DONE	#TX_FAIL	LoEP [%]	LoIP [%]	Av. Pkt Size [B]
Winter	2.84	8.31	691	104	13.08	4.57 × 10^−6^	16.62
Spring	1.83	9.20	1141	72	5.94	5.18 × 10^−10^	18.40
Summer	1.86	9.19	1164	78	6.28	8.96 × 10^−10^	18.38
Autumn	2.84	8.27	676	132	16.34	3.12 × 10^−5^	16.54
Year	2.34	8.74	3672	386	9.51	1.17 × 10^−7^	17.48

**Table 5 sensors-23-06568-t005:** Results obtained per season using the real dataset, with C = 4.5 F and PV = 0.01 m${}^{2}$.

Season	Average *T* [h]	Average NR	#TX_DONE	#TX_FAIL	LoEP [%]	LoIP [%]	Av. Pkt Size [B]
Winter	3.55	7.64	540	127	19.04	3.15 × 10^−4^	15.27
Spring	1.76	9.28	1193	81	6.36	7.83 × 10^−10^	18.56
Summer	1.66	9.27	1291	85	6.18	6.17 × 10^−10^	18.54
Autumn	3.24	7.94	581	138	19.19	2.02 × 10^−4^	15.89
Year	2.55	8.53	3605	431	10.68	5.13 × 10^−7^	17.07

**Table 6 sensors-23-06568-t006:** Capacitor unit characteristics and cost.

Component Type	Manufacturer Code	C [F]	Cost per Unit [€]
A	SCMR22D155PRBB0	1.5	3.77
B	SCMS22D255PRBB0	2.5	2.23
C	SCMT32D755SRBB0	7.5	4.75

**Table 7 sensors-23-06568-t007:** Photovoltaic cell characteristics and cost.

Component Type	Manufacturer Code	Power [W]	Size [m${}^{2}$]	Cost per Unit [€]
A	313070004	0.5	0.0038	3.4
B	186-0599	1.5	0.0263	16
C	914-8445	5	0.06	64.56

**Table 8 sensors-23-06568-t008:** Capacitor cost analysis for LoEP values below 5%.

	A-Type Capacitor			B-Type Capacitor			C-Type Capacitor		
Cap Required [F]	Cap Obtained [F]	Cap Number	Cost [€]	Cap Obtained [F]	Cap Number	Cost [€]	Cap Obtained [F]	Cap Number	Cost [€]
3.0	3.0	2	7.54	5	2	**4.46**	7.5	1	4.75
3.5	4.5	3	11.31	5	2	**4.46**	7.5	1	4.75
4.0	4.5	3	11.31	5	2	**4.46**	7.5	1	4.75
4.5	4.5	3	11.31	5	2	**4.46**	7.5	1	4.75
5	4.5	3	11.31	5	2	**4.46**	7.5	1	4.75
5.5	6	4	15.08	5	2	**4.46**	7.5	1	4.75
6	6	4	15.08	7.5	3	6.69	7.5	1	**4.75**
6.5	6	4	15.08	7.5	3	6.69	7.5	1	**4.75**
7	7.5	5	18.85	7.5	3	6.69	7.5	1	**4.75**
7.5	7.5	5	18.85	7.5	3	6.69	7.5	1	**4.75**

**Table 9 sensors-23-06568-t009:** Cost analysis of photovoltaic cell for LoEP values below 5%.

	A-Type Photovoltaic Cell			B-Type Photovoltaic Cell			C-Type Photovoltaic Cell		
PV Required [m${}^{2}$]	PV Obtained [m${}^{2}$]	Cells Number	Cost [€]	PV Obtained [m${}^{2}$]	Cells Number	Cost [€]	PV Obtained [m${}^{2}$]	Cells Number	Cost [€]
0.02	0.019	5	17	0.026	1	**16**	0.06	1	64.56
0.025	0.027	7	23.8	0.026	1	**16**	0.06	1	64.56
0.03	0.030	8	**27.2**	0.052	2	32	0.06	1	64.56
0.035	0.034	9	**30.6**	0.052	2	32	0.06	1	64.56
0.04	0.042	11	37.4	0.052	2	**32**	0.06	1	64.56
0.045	0.046	12	40.8	0.052	2	**32**	0.06	1	64.56
0.05	0.050	13	44.2	0.052	2	**32**	0.06	1	64.56
0.055	0.057	15	51	0.052	2	**32**	0.06	1	64.56

## Data Availability

No data available.

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
