# Peer review of "Adaptive Algorithms for Batteryless LoRa-Based Sensors"

_sensors, 2023, doi:10.3390/s23146568_

Round 1

Reviewer 1 Report

The work researched adaptive transmission algorithms to improve the performance of battery-less IoT sensors based on the LoRa protocol, the device power consumption during sensor measurement and/or transmission events is characterize, and different scenarios and tune dynamically the most critical network parameter is considered. However I have some concerns presented to the author for further revision;

1). In the introduction part and related work, the author should conduct an in-depth discussion on the research progress of transmission algorithms of battery-less IoT sensors to highlight why the research on adaptive algorithms of battery-less IoT sensors is important and the main contributions of the paper. At the same time, the sixe contributions should be summarized into 3 points;

2). For section 3.2 adaptive transmission algorithms, it is recommended to add a flowchart to describe the logical framework of the adaptive algorithm in detail;

3). In the simulation and evaluation, the source of the real dataset should be given in detail; and the calculation time and stability of the adaptive transmission algorithm should be presented in a statistical table;

4) The conclusion should be improved. The conclusion should not simply repeat the research content of the thesis, but should conduct an in-depth analysis of the obtained results, so as to provide research basis and inspiration for later readers and scholars;

5) There are some grammatical errors and expressions in the paper that need further improvement;

Reviewer 2 Report

In this work, the authors analyzed LoRa-based batteryless devices, optimizing the transmission parameters through an energy-aware adaptation algorithm. By characterizing and modeling the power consumption of two battery less sensors and using  theoretical energy models and real radiation traces, they study how to dynamically tune the transmission interval and the payload size to with the available energy. Then, they develop a general approach for designing  capacity-based storage and PV panel size, obtaining a cost-effective methodology to design battery-less solutions for ambient energy-powered LoRa sensors. The results showed that, for a give threshold probability of power failure the proposed approach can optimize the energy consumption of the device automatically setting the relevant transmission parameters.  The work deserves to be published.

The work is well done and well written, there a need of small writing corrections like the one in line 60, where it seems to have a word missing. Attached is a file were some indications are made, which includes the missing word on line 60 and some suggestions that could improve the clarity of text.

Reviewer 3 Report

Review Report

Manuscript

Adaptive Algorithms for Batteryless LoRa-based Sensors

Manuscript # sensor 2452377

Submitted in the journal “Sensors

Summary

This manuscript focuses on the adaptive transmission algorithm for enhancing the performance of battery-less IoT devices, focusing on the packet size of LoRa protocol. packet size, along with the energy management for the transmission system. This may help in the utilization of battery-less IoT devices. This manuscript is well written, however, its publications demand serious attention, following are my comments;   

Comments

·       Main concern for the manuscript lack of a real experimental picture as proof, there is not a single picture of experimentation of used devices TTGO and FiPy.

·       This creates serious doubt about the originality of the manuscript, as storytelling and writing is always easy task.

·       Because of no experimental picture it is hard to understand whether it is an experimental study or a simulation-based study.

·       Devices have only been introduced while their explanation and real work are missing.

Suggestions

·       Caption of table always come before table (please check this for all the tables)

·       Line 52 (Latter)??

·       Line 478 “the result showed that for a give ………..”

·       Line 4 “studied”.

English is fine except for some typo

Reviewer 4 Report

The article is on a very good level.

I have got only a few remarks.

Verify, whether table descriptions should not be above the table itself, therefore the opposite of Figure descriptions, which are found below the given Figure. 

Consider enlarging axis descriptions in Figures 8 and 9, and try to unify the size of all such descriptions throughout every Figure.

Some minor English adjustments would be beneficial, for example on line 477: The results showed that, for a GIVEN threshold probability of power failure,...
